# A 120-ke^−^ Full-Well Capacity 160-µV/e^−^ Conversion Gain 2.8-µm Backside-Illuminated Pixel with a Lateral Overflow Integration Capacitor [note 1]

**DOI:** 10.3390/s19245572

**Published:** 2019-12-17

**Authors:** Isao Takayanagi, Ken Miyauchi, Shunsuke Okura, Kazuya Mori, Junichi Nakamura, Shigetoshi Sugawa

**Affiliations:** 1Brillnics Japan Inc., 6-21-12 Minami-Oi, Shinagawa-ku, Tokyo 140-0013, Japan; miyauchi.ken@brillnics.com (K.M.); mori.kazuya@brillnics.com (K.M.); nakamura.junichi@brillnics.com (J.N.); 2Department of Science and Engineering, Ritsumeikan University, 1-1-1, Nojihigashi, Kusatsu, Shiga 525-8577, Japan; sokura@fc.ritsumei.ac.jp; 3Graduate School of Engineering, Tohoku University, 6-6-11-811, Aza-Aoba, Aramaki, Aoba-ku, Sendai, Miyagi 980-8579, Japan; shigetoshi.sugawa.d4@tohoku.ac.jp

**Keywords:** CMOS image sensor, lateral overflow integration capacitor, wide dynamic range, high sensitivity, high full-well capacity, single exposure, backside illumination

## Abstract

In this paper, a prototype complementary metal-oxide-semiconductor (CMOS) image sensor with a 2.8-μm backside-illuminated (BSI) pixel with a lateral overflow integration capacitor (LOFIC) architecture is presented. The pixel was capable of a high conversion gain readout with 160 μV/e^−^ for low light signals while a large full-well capacity of 120 ke^−^ was obtained for high light signals. The combination of LOFIC and the BSI technology allowed for high optical performance without degradation caused by extra devices for the LOFIC structure. The sensor realized a 70% peak quantum efficiency with a normal (no anti-reflection coating) cover glass and a 91% angular response at ±20° incident light. This 2.8-μm pixel is potentially capable of higher than 100 dB dynamic range imaging in a pure single exposure operation.

## 1. Introduction

High dynamic range (HDR) imaging technology is widely demanded and has been introduced in many applications. For example, a sensor dynamic range (DR) of more than 85 dB is required to cover the human face and a traffic sign at night in the same scene. Since the DR of usual complementary metal-oxide-semiconductor (CMOS) image sensors (CISs) is about 70 dB or less, many approaches have been proposed to enhance the DR performance. Table 1 shows representative approaches for HDR CISs. HDR schemes are basically categorized into linear response approaches and non-linear response approaches. Nonlinear response approaches include logarithmic compression, knee compression, timestamp conversion, pulse modulation, etc., and some combinations of these approaches. The key advantage of the non-linear response approaches is the lower interface cost because the HDR signal is compressed at the early signal detecting stage, then read out with a reduced data bandwidth.

In contrast, the linear response approaches, as represented by multiple exposure HDR schemes, treat data in a linear photoconversion manner. The multiple linear response signals are combined to obtain an HDR signal. The linear response signal is preferred for image processing or data analysis because of signal integrity. Linear response approaches are also classified into multiple exposure HDR (MEHDR) schemes and single exposure HDR (SEHDR) schemes. In recent trends, the SEHDR approaches have been highlighted because of less motion artifacts and less LED light flicker problems than the MEHDR.

The authors reported an over 87 dB SEHDR CIS with a 3.0 μm pixel introducing a high full-well capacity (FWC) photodiode (PD) and a multiple gain pixel technology [1,2,3,4]. In the previous sensor, photoelectrons accumulated in the photodiode are read out two times in different gains, then combined into an HDR signal. The CIS successfully demonstrated the benefits of the SEHDR, such as no motion artifacts; however, it is difficult to drastically increase the DR further because of the physical limitations of the FWC of pinned PDs.

To break through the restriction and to obtain higher SHEDR performance than that of the previous sensor, lateral overflow integration capacitor (LOFIC) technology [5,6,7,8,9] is introduced with backside illumination image sensor (BSI) technology. A basic pixel schematic and the operation timing of the LOFIC pixel are explained in Section 2 and the design of the BSI LOFIC pixel is described in Section 3. Fabrication and measurement results and sample images are shown in Section 4, and then the conclusion is given in Section 5.

## 2. Lateral Overflow Integration Capacitor (LOFIC) Pixel

Figure 1 shows a circuit diagram of the LOFIC pixel. Compared to a usual 4-transistor-type pixel configuration, an overflow charge storage gate (SG) and a charge storage capacitor (CS) are implemented between a floating diffusion node (FD) and a reset gate (R). Wherein, the electric potential heights of the TG and the SG during the exposure period are designed such that once the photodiode is saturated, overflow charge from the photodiode (PD) flows into the FD node and is accumulated in a capacitor of the FD (CFD). Furthermore, when the FD node gets saturated, the excess charge flows into the CS capacitor through the SG gate. The LOFIC pixel is capable of accumulating the signal charge at the CFD and the CS even after saturation of the PD.

The LOFIC pixel operation timing and electric potential diagram are shown in Figure 2a,b, respectively. First, SG, R, and TG are all turned ON to reset the PD, the CFD, and the CS; then, the pixel is initialized and exposure starts at time t1. When the exposure is completed, an SEL pulse is applied to a select transistor and the first initial revel (HCG RST) is sampled at a time t3. After the charge transfer pulse (TG) is applied at a time t3, the first signal level (HCG SIG) is sampled at a time t4. By performing the correlated double sampling (CDS), a high gain and low noise signal readout is obtained in the same manner as the 4 T pixel when the signal charge is lower than the charge storage capacity of the PD. In this condition, the readout conversion gain is also the same as the 4 T pixel and expressed as q/CFD, where q is the elementary charge.

When the highlight signal is illuminated and PD is saturated, the overflow charge from the PD is accumulated in CFD and/or CS. When SG is turned on, the total overflow charge in the FD and the S are merged. The second TG pulse is given at a time t5, and the remaining charge in the PD is also added. Then, the output signal is sampled at a time t6 as a second signal (LCG SIG). After the FD node is reset by R, second reset level (LCG RST) is sampled. After subtracting LCG RST from the LCG SIG, the overall signal charge in the FD and the CS is read out with a low conversion gain as q/(CFD+CS). The second low gain readout is basically the same as the 3 T pixel operation. Readout noise in the low gain condition is determined by a thermal reset noise of the capacitors CFD and CS, and is approximately expressed as $\sqrt{2k_{B}T\left(\mathrm{CFD}+\mathrm{CS}\right)}/q$ (e-rms). Although this noise is larger than that of the high-gain readout, the noise can be neglected when the optical shot noise at the signal conjunction point is sufficiently larger.

While the low-noise readout capability is obtained for low-light signals, the FWC is expanded by the CFD and CS regardless of the charge storage capacity of the PD. Also, both high-gain and low-gain signals have linear photoconversion characteristics, such that signal linearization is processed simply and accurately. In addition, it mitigates the photodiode full-well requirement because FWC can be expanded by the CFD and CS, which allows for significant benefits in photodiode dark current reduction, operating voltage reduction, and pixel operation speed without image lag.

However, the conventional LOFIC pixel still has a disadvantage, which is the low optical performance compared to the typical 4 T pixel because the optical fill factor of the pixel is reduced by the area for the SG and CS. This issue is more severe when the pixel size is reduced.

## 3. Backside Illumination LOFIC Pixel

In order to improve the optical performance, we have introduced a BSI process for a 2.8-μm LOFIC pixel. Figure 3a,b shows conceptual cross-sections of a conventional frontside illumination (FSI) LOFIC pixel and a BSI LOFIC pixel, respectively. In the FSI LOFIC pixel in Figure 3a, light is induced from the top side of the figure such that the PD fill factor is reduced by the CS and other transistors. Reduction of the PD fill factor affects the key optical performance in terms of quantum efficiency or angle dependence, especially for a in small pixel size, meaning there is a fundamental trade-off between the optical performance and dynamic range enhancement by a larger CS.

On the other hand, in the case of a BSI structure shown in Figure 3b, the trade-off problem is resolved because the input light is induced from the backside. There is a deep *N*(*N*^−^) layer (PD region) used to expand the photo conversion layer below CS and other transistors. Then, optically generated electrons flow into an N-type storage diode and are accumulated. Compared to the FSI, benefits of the deep N-layer expanded under the CS are a higher quantum efficiency, lower pixel-to-pixel cross-talk and higher immunity against the ray angle because of a narrower deep p-well (PW) isolation at the pixel boundary. To collect the photogenerated electrons efficiently, the dopant concentration has a gradient from the frontside to the backside, which generates an additional electric field from the backside to the storage diode.

Another structural feature of the pixel is a buried overflow structure beneath the TG and the SG gate. Because of the buried structure, the overflow paths are active while a negative gate pulse is applied to the TG and SG, and the surface potential of the gate is pinned for dark current reduction.

Another buried P layer is implemented between the buried overflow path and the deep N- photoconversion area. The buried P layer acts as a charge diffusion barrier to the FD and the CS node from the PD, suppressing parasitic sensitivity.

The pixel layout configuration of the 2.8-μm BSI LOFIC pixel is shown in Figure 4. A yellow dotted line and black dotted line show pixel boundary and the deep N-region boundary, respectively. The deep N-region is expanded to beneath the CS of a neighboring pixel. While the storage diode locates the right side within the pixel, the deep n-layer keeps the rectangular and symmetric shape in both the horizontal and vertical directions. PN junction isolation is used to surround the FD and SG node to reduce leakage current at these nodes.

Figure 5 is an electric potential cross-sectional view obtained by a 3-D device simulator from the PD to the SG through the TG and the SG regions. The path is also denoted as X–X’ in Figure 4. The electric potential is the initial condition just after the reset operation and negative voltages are applied to the TG and SG gates. The deep N-region is expanded below the CS of the neighboring pixel. The buried overflow paths are formed under the TG and SG gates. The electric potential height at the overflow path is designed to be sufficiently lower than that of the pixel-to-pixel isolation such that the overflow charge from the PD is accumulated in the FD node and the CS node without blooming between pixels due to carrier diffusion.

## 4. Fabrication and Characterization Results

The BSI LOFIC pixel was tested with a prototype sensor and fabricated in a 2.8 V, 65-nm BSI CIS process. The peripheral circuit of the prototype sensor consisted of a pixel array, an on-chip timing generator, a single-slope column analog-to-digital converter array, a 1280 H × 504 V effective pixel array, and a 1280 H × 216 V sub array with pixel layout variations.

Figure 6 shows the chip microphotograph. The chip was assembled in a 64-pin plastic-leaded-chip-carrier (PLCC) package with a standard cover glass that had a transmittance of about 93%.

In order to compare the performance between the HCG mode and the LCG mode, their characteristics were measured and analyzed as separately as possible. Photoconversion data obtained in the HCG mode and in the LCG mode are plotted in Figure 7. The horizontal axis and vertical axis show the integration time under a constant light and the number of signal electrons, respectively. The number of signal electrons was estimated from the optical shot noise. Both the HCG and LCG operation modes showed good linear characteristics. The LFW of the HCG mode and LCG mode were obtained as 7 ke^−^ and 120 ke^−^, respectively. While conversion gain of the HCG mode of 160 μV/e^−^ is comparable with our previous 3.0-μm CIS, the obtained LFW of 120 ke^−^ in the LCG mode was about 3 times larger than that of the previous sensor of about 40 ke^−^.

Next, the spectral response of the quantum efficiency (QE) is shown in Figure 8. The measured peak for QE was 70% with a non-anti-reflection coating cover glass. Considering the effective transmittance of the glass lid was 93%, the intrinsic peak QE was predicted to be about 75%. Although the value was slightly lower than the previous sensor, there was no significant degradation of QE with the BSI LOFIC structure.

Spectral response of the QE in the LCG mode was almost the same as that of the HCG mode, which is very important to prevent color noise when the two signals are combined for linearization. It suggests that the parasitic sensitivity at the FD node or SG node was negligibly small. It also suggests that the overflow charge was completely transferred to the FD node or SG node. If the charge overflow to the neighboring pixel occurred when photodiode was saturated in the LCG mode, it should affect the color cross-talk, and thus produce a discrepancy in the spectral response.

Figure 9 shows the ray angle dependence of QE in (a) the vertical direction and (b) the horizontal direction without a solid angle correction of 1/cosθ, where θ is defined as angle from the perpendicular incidence. As expected from the layout symmetry of the deep N^−^ region shown in Figure 4, the angle dependence in the horizontal direction and vertical direction were almost the same. Photosensitivity at ±20° was higher than 90%, thus the sensitivity degradation at ±20° was lower than 10% of that of perpendicular light.

To evaluate the immunity of the pixel isolation performance against the ray angle, the angle dependence of the Gr/Gb sensitivity ratio was measured. If there is significant cross-talk and asymmetry, the angle dependence of the Gr/Gb sensitivity ratio should show a significant difference in the rotating direction and/or input color dependence. Figure 10a,b shows the measured results of Gr/Gb ratio in the vertical rotation and the horizontal rotation, respectively. When a blue light condition is referred to, a less-than-2% deviation was achieved, which suggests that more than 98% of photoelectrons generated in the BSI side under the CS were collected in the FD without crosstalk to the neighboring pixels.

Figure 11 shows the measurement result of the FD dark current histogram at 60 °C. We compared the PN-junction isolated FD and shallow-trench isolated FD using pixel layout variations in the test chip. As a result, the dark current of the PN-junction isolated FD was one-tenth smaller than that of the shallow-trench isolated FD. With tuning of the concentration of n+ in the FD, further reduction of the dark current is expected.

Figure 12 shows HDR sample images obtained using this prototype chip. We captured some color objects that were illuminated by a high-illuminance halogen light or a low-illuminance miniature light. The ratio of the illuminance between them was about 70 dB. This prototype chip successfully reproduced an SEHDR color image without halation.

## 5. Conclusions

Performance comparison of several SEHDR image sensors [7,8,9,10,11] with the presented pixel is summarized in Table 2. We have achieved a 120 ke^−^ FWC and a 160 µV/e^−^ conversion gain, C.G. with the 2.8-µm non-shared LOFIC pixel. A 75% peak QE to the perpendicular light and over 91% AR at ±20° without blooming were obtained. Supposing that the pixel is implemented in the same sensor platform of the previous sensor, over a 100 dB DR in the SEHDR mode is expected.

## Figures and Tables

**Figure 1 sensors-19-05572-f001:**
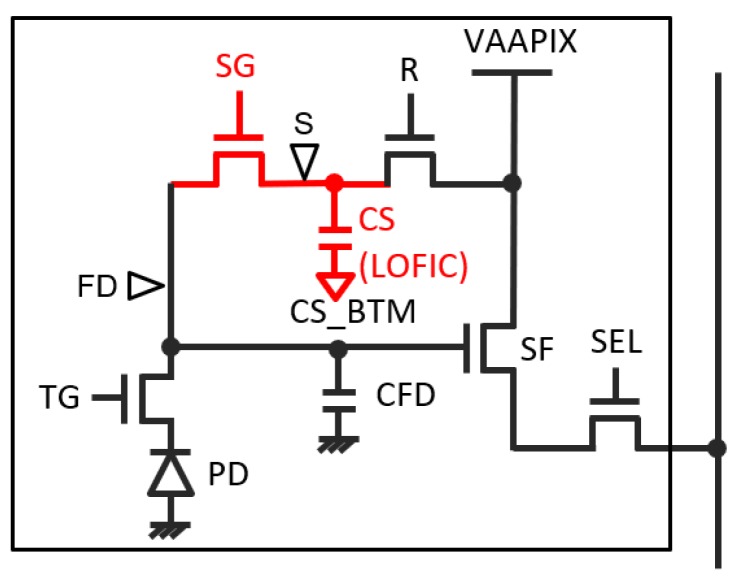
Pixel circuit schematic of a lateral overflow integration capacitor (LOFIC) pixel.

**Figure 2 sensors-19-05572-f002:**
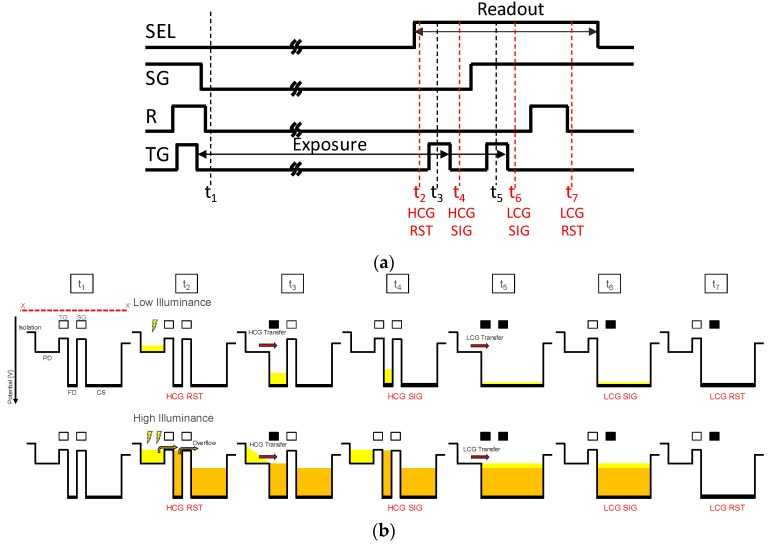
(**a**) Pixel operation timing of LOFIC CISs. (**b**) Potential diagram of LOFIC CISs. There are two charge transfer processes, high conversion gain (HCG) charge transfer and log conversion gain (LCG) charge transfer. SEL, SG, R and TG are gate control pulse. RST and SIG denote sample-and-hold timings for the reset level and the signal revel, respectively.

**Figure 3 sensors-19-05572-f003:**
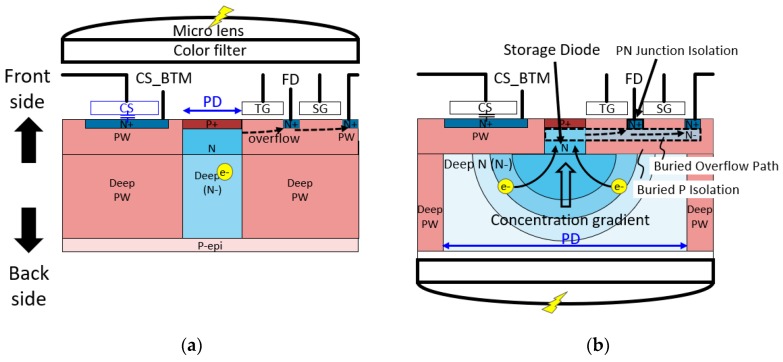
Schematic cross-sectional view of (**a**) a conventional FSI LOFIC pixel and (**b**) the proposed backside-illuminated (BSI) LOFIC pixel. CS_BTM is bottom electrode of the capacitor CS.

**Figure 4 sensors-19-05572-f004:**
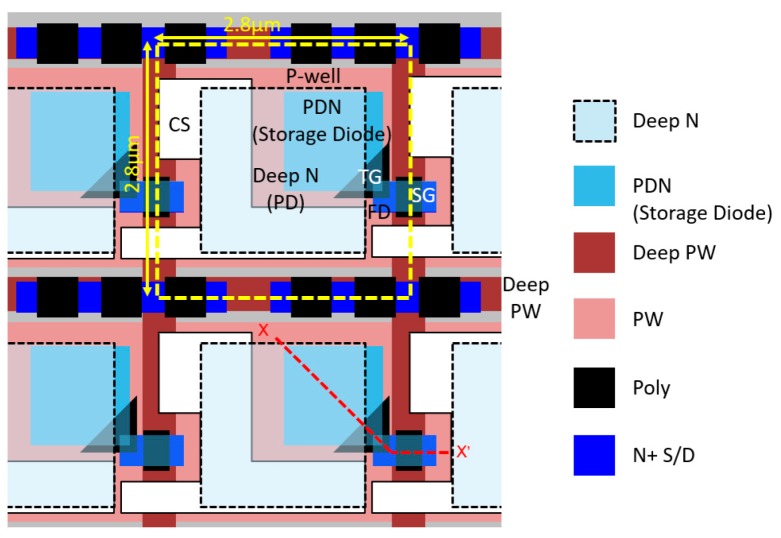
Schematic layout configuration of the 2.8-μm BSI LOFIC pixel. PDN and S/D show photodiode N implantation and source/drain implantation for transistors, respectively.

**Figure 5 sensors-19-05572-f005:**
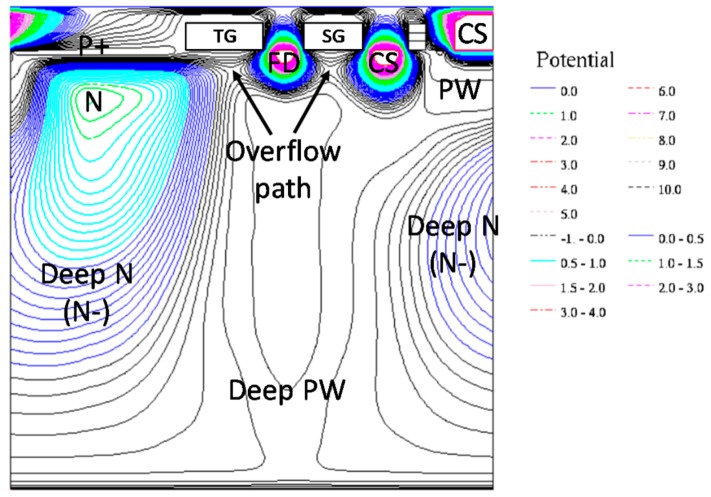
Potential distribution.

**Figure 6 sensors-19-05572-f006:**
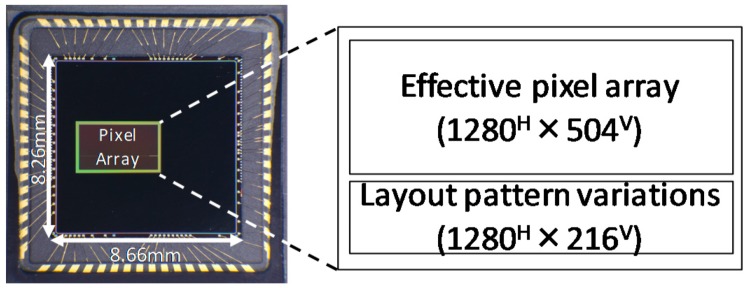
Chip photograph. Assembled in a 64 pin PLCC package.

**Figure 7 sensors-19-05572-f007:**
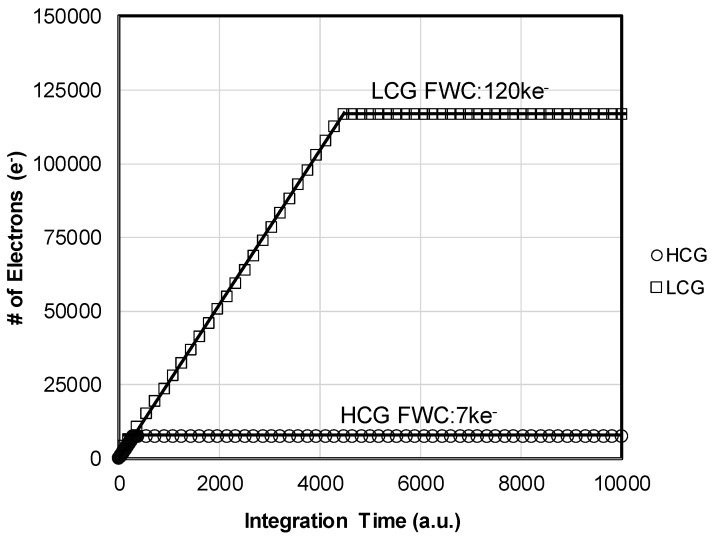
Photo conversion characteristics measured in the HCG and LCG modes. FWC: full-well capacity.

**Figure 8 sensors-19-05572-f008:**
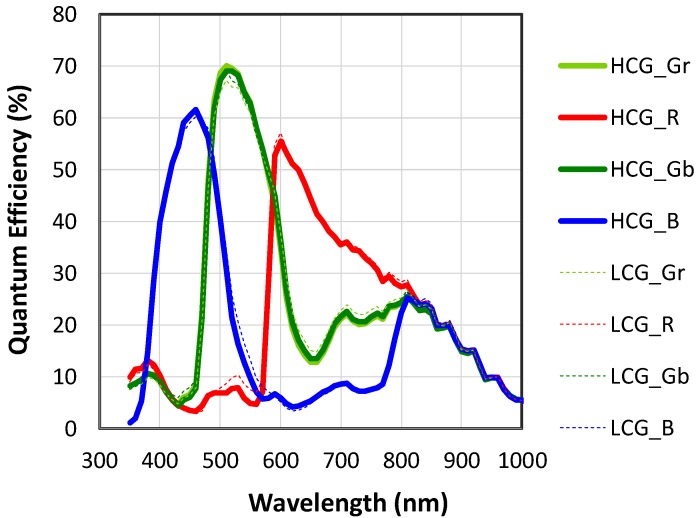
Quantum efficiency measured in HCG and LCG modes.

**Figure 9 sensors-19-05572-f009:**
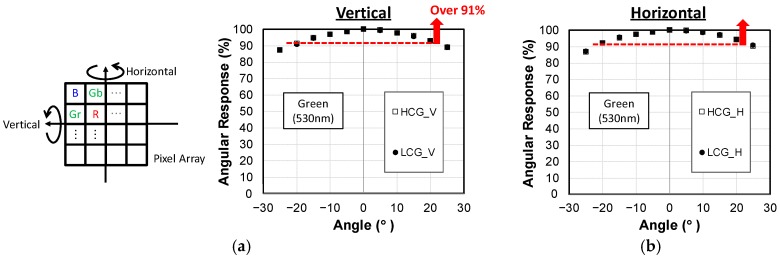
Angular response measured in HCG and LCG modes. (**a**) Vertically rotated; (**b**) Horizontally rotated.

**Figure 10 sensors-19-05572-f010:**
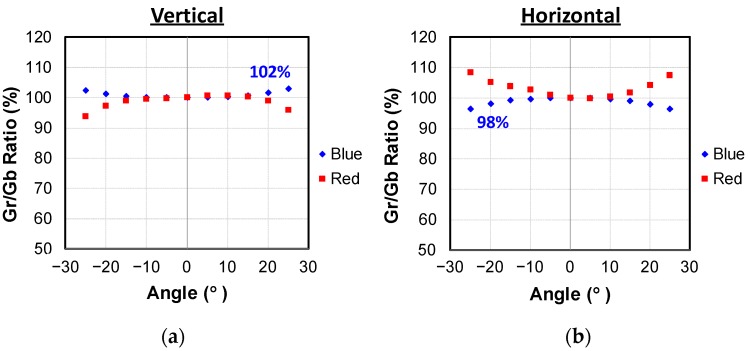
Gr/Gb ratio. (**a**) Vertically rotated; (**b**) Horizontally rotated.

**Figure 11 sensors-19-05572-f011:**
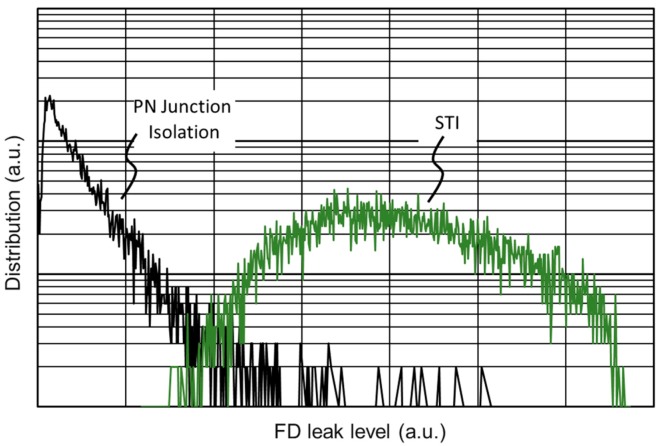
Floating diffusion node (FD) dark current histogram at 60 °C comparing between the shallow-trench isolation (STI) and PN-junction isolation.

**Figure 12 sensors-19-05572-f012:**
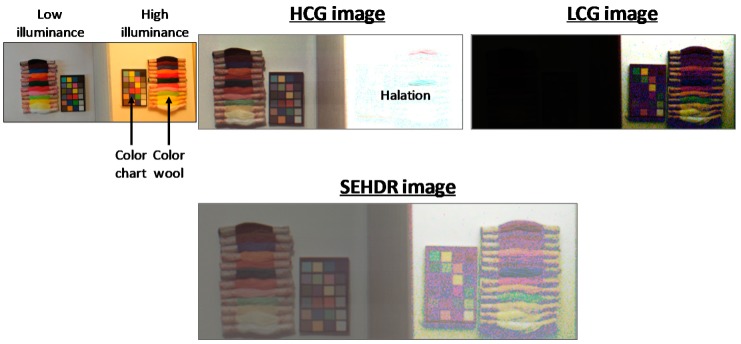
Sample images captured by HCG, LCG, and single-exposure high dynamic range (SEHDR) mode.

**Table 1 sensors-19-05572-t001:** Representative approaches to enhance the dynamic range of CMOS image sensors.

Signal Response	Nonlinear Response	Linear Response	
Schemes	Logarithmic compression Knee compression Timestamp conversion Light to frequency conversion	Multiple Exposure	Single Exposure
		Multiple frames with different integration times Line or pixel interleave with different integration times	Multiple gain readout Multiple-sensitivity pixel composites

**Table 2 sensors-19-05572-t002:** Comparison of specifications and performances between several HDR CISs.

	[12] This Work	[10] @IEDM2018	[9] @VLSI2017	[8] @VLSI2016	[7] @JJAP	[11] @IEDM2018 Not LOFIC	[1,2,3,4] @IISW2017 Not LOFIC
Process	BSI 65 nm	90 nm	180 nm FEOL/90 nm BEOL	Stack 45 nm pixel/65 nm ASIC	FSI 180 nm	Stack	BSI 65 nm
VAAPIX	2.8 V	2.9 V	3.3 V	3.3 V	N/A	N/A	2.8 V
Pixel size	2.8 µm	3.0 µm	3.875 µm	6.6 µm (4 × 4 sub-pixels)	3.0 µm	1.5 µm	3.0 µm
# of Pixels	1280 × 504	1920 × 1200	1.3 M	640 × 476	1280 × 960	8 M	1928 × 1088
FD/SF Shared	1PD/FD	1PD/FD	1PD/FD	16PD/FD	2PD/FD	2PD/FD	2PD/FD
HCG FWC	7 ke^−^	10 ke^−^	5 ke^−^	N/A	N/A	4.5 ke^−^	N/A
LCG FWC	120 ke^−^	78.5 ke^−^	224 ke^−^	220 ke^−^	69 ke^−^	13 ke^−^	45 ke^−^
FWC (/1 µm^2^)	15.3 ke^−^	8.7 ke^−^	14.9 ke^−^	5.1 ke^−^	7.7 ke^−^	5.8 ke^−^	5.0 ke^−^
HCG-C.G.	160 µV/e^−^	N/A	N/A	N/A	84 µV/e^−^	200 µV/e^−^	152 µV/e^−^
LCG-C.G.	10 µV/e^−^	N/A	N/A	N/A	N/A	20 µV/e^−^	21 µV/e^−^
AR @±20°	91%	N/A	N/A	N/A	N/A	N/A	90.40%
Peak Q.E.	70%	N/A	N/A	N/A	N/A	80%	78.80%

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
