# Peer review of "A 120-ke Full-Well Capacity 160-µV/e Conversion Gain 2.8-µm Backside-Illuminated Pixel with a Lateral Overflow Integration Capacitor"

_sensors, 2019, doi:10.3390/s19245572_

Round 1

Reviewer 1 Report

The paper is well written. It can be accepted after minor revision. 

(1) p. 5, line 6 

  berried --> burried ? 

(2) A possible problem of BSI for the wide dynamic range image sensor using the LOFIC technology is the parasitic sensitivity. If a very strong light is coming to the FD or SG, the high conversion gain signal may saturate or the linear response is deteriorated. A technique for suppressing parasitic sensitivity by the charge-diffusion barrier is mentioned in page 5, but the authors should discuss the problem of the parasitic LIGHT sensitivity of BSI technology. 

(3) The QE of the LOFIC pixel using the BSI technology is slightly smaller than that of the previous sensor (is it a FSI senosr?).  The physical reason why the QE of the BSI-based sensor is slightly smaller than the previous sensor should be desribed. 

Author Response

(1) p. 5, line 6 

  berried --> burried ? 

[Answer] Thank you, we will fix it

(2) A possible problem of BSI for the wide dynamic range image sensor using the LOFIC technology is the parasitic sensitivity. If a very strong light is coming to the FD or SG, the high conversion gain signal may saturate or the linear response is deteriorated. A technique for suppressing parasitic sensitivity by the charge-diffusion barrier is mentioned in page 5, but the authors should discuss the problem of the parasitic LIGHT sensitivity of BSI technology. 

[Answer] The biggest concern was actaully difference between LCG and HCG in their spectral response becasue only LCG mode has additional photo-conversion area aroudn the SG switch.  The measured spectral response didn't show significant difference, so that we concluded that contribution from the parasitic photoconversion area is little.

By the way, under very strong exposure conditions, photodiode is saturated and the overflow charge is integrated in the LOFIC capacitor anyway, then readout the overall charge after summing remaining PD charge and the LOFIC charge without loss of signal charge.  Because we just use the LCG singal for strong light, we don't have to worry about nonlinearity of the HCG mode in high light region.

(3) The QE of the LOFIC pixel using the BSI technology is slightly smaller than that of the previous sensor (is it a FSI senosr?).  The physical reason why the QE of the BSI-based sensor is slightly smaller than the previous sensor should be desribed. 

[Answer] Good question. We think this is because the microlens optimization is not good enough.  We are going to improve it in the next change.

Reviewer 2 Report

typos:

#1: lay angle => ray angle?  pg 4 2nd line from bottom, pg 7 4th line from bottom, pg 8 3rd line from top

#2: berried => buried, pg 5 6th line from top

Author Response

For #1

Yes, it's typo.  We will fix it.

For #2

Yes, it's also typo... We will fix it

Thank you very much